# Relationship between COVID-19 Pandemic Confinement and Worsening or Onset of Depressive Disorders

**DOI:** 10.3390/brainsci13060899

**Published:** 2023-06-02

**Authors:** Daniela Camargo, Elisabet Navarro-Tapia, Jordi Pérez-Tur, Fernando Cardona

**Affiliations:** 1Faculty of Health Sciences, Valencian International University (VIU), 46002 Valencia, Spain; danielacamargoarango95@gmail.com (D.C.); elisabet.navarro@campusviu.es (E.N.-T.); 2Unitat de Genètica Molecular, Instituto de Biomedicina de Valencia-CSIC, 46010 Valencia, Spain; jpereztur@ibv.csic.es; 3Centro de Investigación Biomédica en Red en Enfermedades Neurodegenerativas, Instituto de Salud Carlos III (CIBER-CIBERNED-ISCIII), 28029 Madrid, Spain

**Keywords:** depressive disorder, COVID, lockdown, pandemic

## Abstract

Several studies indicate that the pandemic and associated confinement measures may have had an impact on mental health, producing the onset or persistence of symptoms such as stress, anxiety, depression, and fear. This systematic review aims to identify the factors influencing the onset or worsening of depressive symptoms during COVID-19-related confinement. Our systematic search produced 451 articles from selected databases, 398 of which were excluded based on established criteria, while 53 were selected for review. Most studies have reported an increase in the prevalence of depressive symptoms in the general population during the first weeks of confinement. The predominant risk factors associated with the appearance of depressive symptoms included female sex, low educational level, young age, economic difficulties, comorbidities, and a history of previous depressive episodes. People with a pre-existing diagnosis of depressive disorder generally experienced a worsening of their symptoms during confinement in most of the reviewed studies. Moreover, symptomatology persisted at higher levels post-confinement, without significant improvement despite relief in confinement measures. Therefore, ongoing evaluations of post-pandemic depressive symptoms are necessary to advance the knowledge of the relationship between pandemics and depression, allowing accurate conclusions and associations to be made.

## 1. Introduction

Generally, pandemics are defined as conditions with a high incidence or wide prevalence, often associated with rapid temporal and geographic spread. They are typically caused by infectious diseases such as bacteria, fungi, parasites, or viruses [1]. The latest pandemic, caused by SARS-CoV-2, which is associated with the severe acute respiratory syndrome (SARS), was declared by the WHO on 11 March 2020. Physical symptoms exhibited by patients infected with the novel coronavirus have been extensively studied worldwide. Commonly reported symptoms include fever, cough, and muscle pain, as well as specific manifestations associated with severe pulmonary and neurological conditions [2]. 

Throughout history, many infectious diseases have been declared pandemics, such as smallpox, the Black Death, the Spanish flu, and human immunodeficiency virus (HIV) [1]. These diseases have been extensively researched due to the physical harm they cause to infected individuals, and also due to the potential psychological repercussions for these individuals and their social environment. Evidence suggests that people experienced post-traumatic stress disorder during the Ebola outbreak, and during HIV infection, and this has been associated with a higher prevalence of mental health issues [3].

Research suggests that pandemics have a significant impact on mental health, observed both during and after the pandemic. This impact is typically manifested through symptoms of stress, anxiety, depression, fear, and even psychotic disorders [4,5,6]. Several studies have concluded that during 2020, the COVID-19 pandemic led to a 27.6% increase in cases of major depressive disorders and a 25.6% increase in cases of anxiety disorders worldwide [7]. Long-term neuropsychiatric sequelae of COVID-19 are now recognized as key symptoms of post-acute COVID-19 syndrome, including depressive symptoms, anxiety, and cognitive impairments [8]. Significant depressive symptoms have been reported in approximately 30–40% of patients for up to 12 months after COVID-19 infection [9,10,11]. Additionally, persistent depressive symptoms have been observed in individuals exposed to confinement, regardless of whether they had the virus infection. However, limited evidence exists regarding the trajectory of depressive symptoms and the factors that contribute to their appearance or exacerbation in individuals with a history of depression, as compared to those without prior psychiatric disorders. In this systematic review, we investigate the factors that influence the appearance or worsening of depressive symptoms during COVID-19 confinement.

## 2. Materials and Methods

We performed, following the PRISMA guidelines (Figure 1) [12], a systematic search to identify studies reporting the effects of COVID-19 pandemic confinement on individuals with and without a history of mental illness. Unlike randomized controlled trials, observational studies are not well-indexed; hence, an empirical search strategy was developed using the PubMed and Scopus databases. The sensitivity of the search strategy was confirmed by checking the reference lists of articles selected and ensuring that no relevant articles were omitted. The search was initially completed on 6 February 2023, further updated on 20 February 2023, and finalized on 27 February 2023. No additional articles were found by examining the reference lists of relevant articles and recent reviews.

### 2.1. Sources of Data and Search Terms

The search was performed using the terms “(depressive disorder) AND COVID AND lockdown” in the mentioned databases. The obtained list of articles was reviewed, and only studies/articles that met the following criteria were included: (a) studies that reported original data published in English; (b) studies that described participants with pre-existing mental illness, with or without a control group; (c) exploratory studies of psychiatric symptoms in the general population during the lockdown; (d) studies that provided quantitative data, including rating scale scores or percentages; and (e) studies which evaluated the effects of the pandemic on mental health.

### 2.2. Exlusion Criteria

Studies were excluded if they did not meet the inclusion criteria and, more specifically, if they met the following criteria: (a) only described mental health in general without characterizing depressive symptomatology; (b) focused on specific populations such as children, adolescents, pregnant women, or patients with specific comorbidities; (c) were editorials, letters, comments, article analyses, case reports, systematic or literature reviews, or meta-analyses; or (d) were studies in hospitalized patients, as it is not clear whether the symptoms were due to confinement or hospitalization.

### 2.3. Data Extraction

The systematic review yielded a total of 53 studies that investigated the research topic, with the main conclusions summarized in Table 1. Data were extracted from the included studies using a form prepared by us. The data extraction form included the following elements: country of origin, sex distribution, and classification of the depressive disorders as minimal/mild or moderate/severe, together with degree of prevalence. This classification of the severity of the symptoms was obtained based on the score of the scales, and it was defined and validated for all scales used by different studies. Furthermore, factors identified as risk-based or protective were extracted from these studies (Table 2). After all the filtering steps, 53 studies were included in this systematic review (Figure 1).

### 2.4. Data Synthesis

This systematic review involved a synthesis of all included studies, with a focus on the key findings in order to address the review’s questions. The review provides details on the country where each study was conducted, the gender distribution, factors that influenced or protected against depressive symptoms during confinement, and other notable relevant findings. In addition, it describes the available evidence supporting findings in people with pre-existing mental illnesses and changes in depressive symptoms after the end of confinement, as well as the classification of the severity of depressive symptoms. The findings presented primarily relate to depressive symptoms during COVID-19 pandemic confinement. Findings relating to other psychiatric symptoms, such as anxiety, sleep disorders, eating disorders, or substance use were not considered.

## 3. Results

The systematic search initially yielded 451 articles. After excluding 100 duplicates and 14 articles published in Q3 and Q4 journals, the remaining 337 articles were screened by reading the abstracts, excluding 261 of them. Out of the remaining 76 articles, 24 were excluded based on the exclusion criteria, leaving the remaining 53 full-text articles for further evaluation. The main conclusions of these studies and a description of the included studies can be found in Table 1 and Table 2, respectively. The studies included were observational, cross-sectional or longitudinal, and were conducted in various countries, specifically: Italy (17%), Germany, Spain (9.4% each), Austria, the UK (7.5% each), France, Holland (5.7% each), India, Libya (3.8% each), Argentina, Australia, Chile, England, Greece, Ireland, Kuwait, Netherlands, Nigeria, Poland, Portugal, Qatar, South Africa, and the USA (1.8% each). Table 1 summarises the main conclusions of the studies included in this systematic review.

Regarding the onset or exacerbation of depressive symptoms during confinement, most of the reviewed studies conclude that they increased during confinement [14,15,16,17,18,19,20,21,22,24,25,27,28,29,30,31,32,35,36,37,38,39,40,43,45,46,47,48,50,52,53,56,57,59,61,62,63,64,65]. However, some studies conclude that this increase occurred only at the beginning of the confinement [13,33,34,42,60], whereas others support a decrease [23,26,41,44] or no change in symptoms [49,51,54,55,58]. In one study the prevalence of depressive symptoms was higher during confinement (24.7%) compared to pre-pandemic levels [14]. Other works found that depressive symptoms were found to have increased in 12.9% [27] to 26% [30] of participants, with no significant associations with sex, age, or pre-existing mental health problems. Two studies indicated that individuals with a pre-pandemic depression diagnosis experienced higher level of depressive symptoms in 2020 and 2021, while depressive symptoms progressively increased between 2017 and 2020, except among individuals over 65 and African Americans [28,29]. In another study, the prevalence of major depressive disorders decreased significantly between February 2019 and April 2020, remaining relatively stable during the six weeks of confinement [26]. One study found that diagnoses of depressive disorder decreased during the initial confinement period but slightly rebounded afterward, with higher diagnoses among older individuals during confinement and younger individuals at the end of confinement [23]. Additionally, in other studies, the psychological impact increased at the beginning of the lockdown but gradually decreased afterward, with higher prevalence one year after confinement began [24,34]. 

On the other hand, it has also been reported that depressive symptoms were more frequent and severe among patients with a history of depressive disorder, with boredom and changes in psychotropic drug use predicting moderate to severe depression in this group [13,19,21,28,31,34,39,42,48,50,52,54,56,58,61,64]. Moreover, during the initial stages of confinement, a significant proportion of individuals without a history of depression (26%) and with a history of depression (61%) exhibited moderate to severe depressive symptoms, which significantly decreased after 20 weeks [13].

Regarding risk and protective factors for depressive symptoms, one study found associations in specific age groups with several factors, such as sex, education level, continuity of income, financial situation, and health problems [20]. Another study identified detachment and low levels of social support as strongly associated factors [19], while the perceived risk of COVID-19 was found to increase the probability of depressive symptoms, with a 1.93-fold increase per unit increase in perceived risk [21]. Compliance with isolation measures was linked to a higher likelihood of depressive symptoms [32]. Engaging in sports, housework, or having social interactions helped participants feel more relaxed [25]. Being married was associated with a lower risk of depression among individuals with mental disorders [24]. Other factors, such as marital status and geographical location, were associated with the severity of depressive symptoms, with individuals living with their partners experiencing better mental health [15,61].

In terms of age and gender, gender was not significantly associated with depressive symptoms in one study [32] but emerged as an important factor in most others [17,22,25,33]. Depressive symptoms varied by age, with higher scores among younger individuals at the beginning of the pandemic [33]. Age was also associated with higher resilience, partially mediating the impact of stressful events on depressive symptoms [22].

Regarding the variability of symptoms, participants commonly reported feelings of worry, isolation, and concerns about the well-being of themselves and their family members during the lockdown [16,46,47]. Depressive symptoms were associated with worsening memory and cognitive functions, particularly among young adults aged 18–26 [17]. Depressed individuals exhibited longer sleep duration, increased time spent in bed, higher consumption of sugary foods, alcohol, and drugs, as well as higher levels of addiction to television, social media, and gaming [18].

Taking these results into account, it is possible to group the most important risk and protective factors, as well as classify symptoms based on studies. This allows for drawing conclusions about the most frequently found outcomes in different studies and deriving new insights in order to better understand the progression of the disease during pandemic confinement. The classification of severity of the symptoms, as well as the risk and protective factors found, were organized in Table 2.

### 3.1. Risk Factors for Depressive Symptoms during Confinement

Twenty-nine studies (54.7%) pointed to female sex as the main risk factor for developing depressive symptoms during COVID-19 pandemic confinement, regardless of the presence of pre-existing mental disorders. Additionally, 25 studies (47.2%) have found that a younger age or being a minor are important factors that predispose the general population to the onset of depressive symptoms. Other socio-demographic factors that have been repeatedly highlighted as predisposing factors to these disorders included low educational level (six studies, 11.3%), economic concerns or low income (thirteen studies, 24.5%), living alone or loneliness (thirteen studies, 24.5%), job loss or being unemployed during the pandemic (nine studies, 17.0%), and fear of COVID-19 (nine studies, 17.0%). Moreover, 15 studies (28.3%) have shown that individuals with a history of depression or related pathologies were at a higher risk of recurrence of depressive symptoms during confinement.

### 3.2. Protective Factors for Depressive Symptoms during Confinement

The most frequent protective factors found against depressive symptoms during COVID-19 confinement were sharing housing with at least one other person, mentioned in eleven articles (20.8%), and engaging in physical activity during the lockdown, which was associated as a protective factor in seven studies (13.2%). Other repeatedly mentioned protective factors were being older (eight studies, 15.1%) or retired (four studies, 7.5%), or having a higher educational level (ten studies, 18.9%) or socioeconomic status (two studies, 3.8%). Perceived social support was also an important protective factor against depressive episodes (five studies, 9.4%), with greater importance in individuals with pre-existing diagnosed depressive disorders.

### 3.3. Classification of Depressive Symptoms during Confinement

Fourteen studies (26.4%) classified the severity of depressive symptoms during confinement. Of these, seven classified the symptoms as mild, moderate, or severe. The rest referred only to cases of moderate or severe symptoms. The average proportion of the described population that presented minimal or mild depressive symptoms was 40.3%, while 23.6% of the evaluated population presented moderate or severe depressive symptoms.

## 4. Discussion

The objective of this study was to conduct a systematic review of the existing evidence on the impact of confinement during the COVID-19 pandemic on depressive symptoms among individuals with and without a history of depression. The study also aimed to identify factors that could aggravate or alleviate the onset of these symptoms. Our comprehensive search strategy exclusively included studies addressing these research questions in the general adult population, with a total of 53 studies meeting the inclusion criteria. The results provide important insights into the risk and protective factors associated with depressive symptoms during confinement in the context of the COVID-19 pandemic, shedding light on potential avenues for intervention and support.

One of the prominent risk factors identified in this analysis is female sex. Most of the studies included in this review consistently pointed to female individuals being at a higher risk of developing depressive symptoms during confinement, regardless of pre-existing mental disorders [13,15,16,17,19,20,24,25,34,38,39,40,41,42,45,46,50,52,53,54,55,56,58,59,60,63,64,65]. Another significant risk factor identified is a younger age or being a minor [13,15,17,19,20,22,25,26,29,32,34,35,40,42,45,48,49,50,54,56,58,59,60,63]. The results indicate that younger individuals are more prone to experiencing depressive symptoms during confinement. This could be attributed to several factors, including disruptions in routine, limited social interactions, and increased stress related to remote learning or unemployment among younger individuals. Socio-demographic factors such as low educational level, economic concerns or low income, living alone or experiencing loneliness, job loss or unemployment during the pandemic, and fear of COVID-19 have also been consistently highlighted as predisposing factors to depressive symptoms [13,14,15,16,17,19,20,21,24,25,26,27,31,32,33,35,36,37,38,39,40,41,45,46,47,48,49,52,54,56,57,58,59,60,61,62,63,64]. These factors reflect the broader societal and economic impact of the pandemic, with individuals facing increased financial strain, social isolation, and uncertainty being more susceptible to depressive symptoms. Furthermore, a history of depression or related pathologies emerged as a significant risk factor for the recurrence of depressive symptoms during confinement [13,19,21,28,31,34,39,42,48,50,52,54,56,58,61,64]. Individuals with a prior history of depression may be particularly vulnerable to the stressors associated with confinement, requiring specialized attention and support to prevent relapse and promote resilience.

On the other hand, several protective factors were identified that can help mitigate the risk of developing depressive symptoms during confinement. Sharing housing with at least one other person emerged as a frequent protective factor [14,15,16,24,31,38,40,46,57,64]. This suggests that companionship can play a crucial role in buffering against the negative impact of confinement on mental health. Encouraging social connections, even in limited settings, can contribute to improved well-being. Engaging in physical activity during lockdown was also found to be protective against depressive symptoms [15,18,31,45,50,58,63]. Regular exercise has long been associated with improved mental health, and this finding emphasizes the importance of promoting physical activity, even in constrained environments. Other protective factors identified include being older or retired, having a higher educational level or socioeconomic status, and perceived social support [16,18,20,21,24,32,37,38,40,44,48,54,58,62,63,64]. These factors reflect the importance of stability, resources, and a supportive environment in mitigating the risk of depressive symptoms during confinement.

It is worth noting that the severity of depressive symptoms during confinement varied across the studies included in this analysis. However, a substantial proportion of the population exhibited mild or minimal symptoms, while a significant portion experienced moderate or severe symptoms.

Overall, this study underscores the complex interplay of various risk-based and protective factors influencing depressive symptoms during confinement. The findings provide valuable insights for mental health professionals, policymakers, and researchers to develop targeted interventions and support systems that address the specific needs of vulnerable populations. By identifying and addressing these risk factors and bolstering protective factors, it is possible to mitigate the impact of confinement on mental health and promote better outcomes.

Several studies conclude that the prevalence of depressive symptoms was higher at the beginning of the COVID-19 pandemic confinement [13,14,15,16,17,18,19,20,21,22,24,25,27,28,29,30,31,32,33,34,35,36,37,38,39,40,42,43,45,46,47,48,50,52,53,56,57,59,60,61,62,63,64,65]. However, importantly, a significant number of studies also conclude that, as the months passed during 2020, the prevalence of these symptoms decreased again, and this can vary depending on the stringency of the confinement measures [13,33,34,42,60]. Additionally, it was found repeatedly that individuals with more severe depressive symptoms also had associated compulsive behaviours, such as sleep and eating disorders, alcohol abuse, or excessive use of social media or television [14,18,32,51,57,58].

Several studies included in this systematic review have concluded that individuals with a previous diagnosis of depressive disorder experienced a worsening of symptoms during the COVID-19 pandemic, which persisted at higher levels than pre-pandemic levels, and did not significantly improve with changes in lockdown measures [19,21,28,31,34,39,42,50,52,54,56,58,61,64]. However, these findings cannot be generalized to the entire population with a history of depressive disorders due to the limits of the studies’ designs, regarding the absence of cohort studies with control groups that would allow for comparisons regarding exposure to risk factors and counteracting findings of other studies. For instance, some studies found that individuals with a history of major depressive disorder did not experience significantly increased severity of depressive symptoms during the COVID-19 lockdown [13,27]. In fact, some studies even reported decreased symptomatology during the confinement period [48].

The studies included in this review were mostly distributed in the United Kingdom and European countries such as Italy and Germany, with only three studies in America, two conducted in Latin America [52,56] and one in the United States [29]. These results have the limitations that they do not include studies from the rest of the continents, so nothing can be said about the possible behaviour of these other populations. The studies included were homogeneous in design and consisted of cross-sectional or longitudinal observational studies. Some studies provided detailed information on the nature and severity of depressive symptoms, including whether individuals with a history of depressive disorder were actively symptomatic or in remission at the start of confinement. This is an important consideration, as it enables differentiation between the impact of confinement on individuals without a history of mental disorders and those with a current condition. Therefore, caution should still be exercised when drawing conclusions from this review. Additionally, it is unclear whether many of the studies were sufficiently representative and had the sufficient statistical power to establish valid conclusions. 

Most studies included in this review used online or telephone surveys, which limits the ability to determine the reliability of self-reported psychiatric diagnoses. Additionally, different scales were used for the measurement and classification of depressive symptoms. The most commonly used instrument was the Patient Health Questionnaire-9 (PHQ-9), a depression questionnaire with nine items, based on DSM-IV criteria, that are validated in different parts of the world for the probable diagnosis of a depressive disorder with a cut-off score of 10 or higher, regardless of age [66]. Other questionnaires and scales used were the Beck Depression Inventory, the International Adjustment Disorders Questionnaire and the DASS-21. However, it is important to note that these questionnaires and scales are not diagnostic tools, and results from different studies included in this review may not be fully comparable due to the use of other instruments for the evaluation of depressive symptoms. 

### 4.1. Strengths, Perspectives and Limitations

The study’s primary strength lies in its identification of risk factors potentially implicated in the onset of depressive symptoms during subsequent lockdowns. These factors can be mitigated through the implementation of preventive measures, including sustained support from trained mental health professionals. Particularly, individuals with prior mood disorders, women, young and older adults, and those lacking robust social support networks stand to benefit from such interventions. Additionally, the findings of this study hold predictive value for future statistical trends in the event of subsequent lockdowns. 

One limitation of this study, as previously mentioned, is the lack of cohort and prospective studies with a comparison group. This limitation restricts the establishment of a statistically significant association between depressive symptoms and the risk factors identified. Another limitation is the lack of studies from all the continents, since nothing can be inferred from a limited number of regions, all with different characteristics. The use of telephone surveys and different scales also limits the comparability of symptoms between different studies.

Note that divergent results from different studies are described in this work. For example, some papers report an increase in depressive symptoms during COVID-19 in people with a previous diagnosis of depression, while others report a decrease in symptoms during the period of confinement. The divergent results obtained in different studies about depression and confinement during the COVID-19 pandemic can be attributed to several factors, including variations in study design, sample characteristics, measurement tools, and contextual factors. Cross-sectional studies provide a snapshot of a particular point in time, whereas longitudinal studies are better suited to capture the dynamic nature of mental health during the pandemic. Differences in participant characteristics may also contribute to divergent results, as noted in the results of this study. Factors such as age, gender, socioeconomic status, pre-existing mental health conditions, and access to resources may influence the impact of confinement on depressive symptoms. The assessment tools used to measure depression and related symptoms can differ across studies. Variations in the choice of validated scales, self-report measures, clinical interviews, or diagnostic criteria can lead to different interpretations and results. Differences in the sensitivity and specificity of these instruments can affect the identification and reporting of depressive symptoms. The specific timing and duration of the confinement period can influence the findings. Studies conducted at different stages of the pandemic or during varying lengths of confinement may capture different phases of the mental health impact. Initial phases of the pandemic may have induced more anxiety and uncertainty, while prolonged confinement may have led to feelings of isolation and reduced social support.

It is important to consider these factors when interpreting divergent results. A comprehensive understanding of the complex interactions between individual, social, and environmental factors is crucial to fully comprehending the impact of the pandemic on mental health outcomes like depression.

### 4.2. The Onset of Depressive Symptoms during Confinement

The first research question aimed to investigate the factors that contributed to the development of depressive symptoms in adults without a prior history of psychiatric illness. Several studies were analysed which examined the impact of the pandemic on mental health in this population group. Most of the studies indicated an increase in the prevalence of depressive symptoms in the general population during the initial weeks of confinement. The risk factors for the onset of depressive symptoms were found to be female sex, low educational level, young age, economic difficulties, comorbidities, and a previous history of depressive episodes (without a diagnosis of chronic depressive disorder). These findings are consistent with previous research conducted worldwide on the risk factors for depressive disorders, although not all studies have found strong associations [67]. 

Furthermore, a significant proportion of the studies included in this review linked the onset of depressive symptoms with pandemic-related changes in regular life, such as having fear of COVID-19 infection, an infected family member, feelings of loneliness, or decreased physical activity. It is worth noting that the studies included in the review did not provide a consensus on the persistence of depressive symptomatology after the disappearance of confinement measures. While some studies reported a decrease in depression levels to pre-pandemic levels, others reported a persistent increase in the prevalence and incidence of these symptoms even a year after the end of confinement.

### 4.3. Worsening of Depressive Symptoms

This systematic review aimed to investigate whether individuals with pre-existing mental illnesses experienced an increase in depressive symptoms during the COVID-19 pandemic lockdown. Most of these studies concluded that individuals with a previous diagnosis of depressive disorder experienced worsening symptoms, which remained at higher levels than pre-pandemic levels and did not show significant improvements with changes in lockdown measures. These findings are in accordance with previous systematic reviews and meta-analyses conducted on this population [68,69]. However, it is important to note that these results cannot be generalized to the entire population with a history of depressive disorders due to the limitations of our study and contradictory findings from other studies. Some studies suggest that individuals with a history of major depressive disorder did not experience a significant increase in depressive symptoms during the COVID-19 pandemic lockdown, and some even reported a decrease [27,41].

Studies in populations with a history of previous depressive disorders consistently show that social support is a protective factor against the worsening of depressive symptoms [19]. This suggests that, given the conditions of isolation during the COVID-19 pandemic that limited social contact and public outings, an increase in symptomatology can be expected in these populations. However, further research is needed to study the course of depressive symptoms after the relief of lockdown measures in people with psychiatric histories, in order to draw more accurate conclusions.

## 5. Conclusions

In conclusion, the studies reviewed suggest that individuals without a history of psychiatric disorders, as well as those with a history of depressive disorder, may experience clinically significant depressive symptoms during pandemic confinement. Other risk factors that showed a significant association with depressive symptoms during the COVID-19 lockdown included female gender, low socioeconomic status, low educational level, economic concerns, single status, and feelings of loneliness. However, the prevalence and severity of depressive symptoms vary across studies, and there are discrepancies in the associated risk and protective factors. Further investigation is needed to better understand the impact of the pandemic and confinement measures on mental health and to develop appropriate prevention and treatment strategies to minimize their negative effects.

## Figures and Tables

**Figure 1 brainsci-13-00899-f001:**
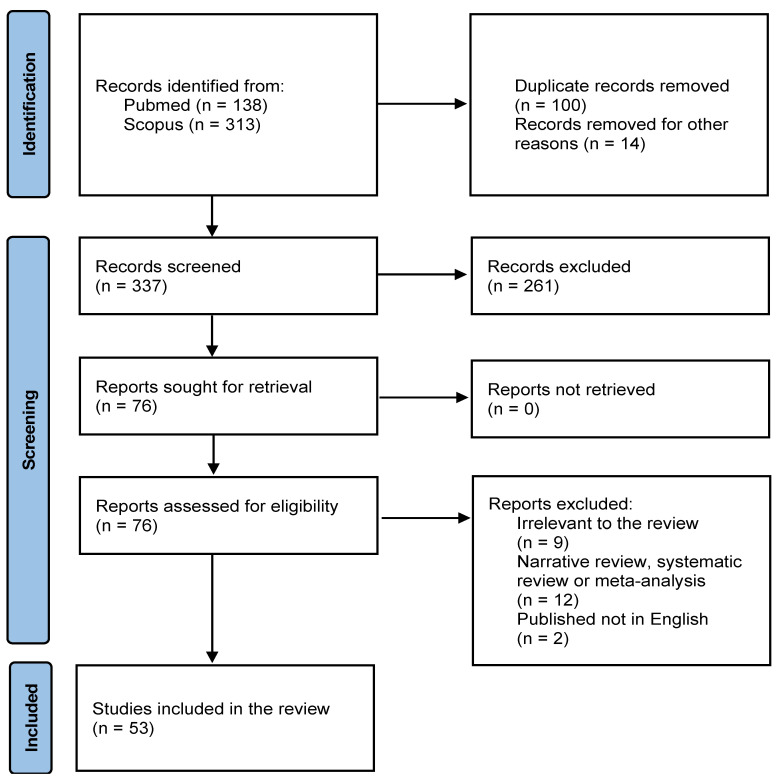
PRISMA flow diagram of the review.

**Table 1 brainsci-13-00899-t001:** Summary of the main findings of the studies included in the systematic review.

Ref.	Summary of Findings
[13]	At the initial stages of the confinement, 26% and 61% of individuals, without and with a history of depression, respectively, presented moderate or severe depression. A significant decrease in depression symptoms was observed after 20 weeks.
[14]	The prevalence of depression symptoms was higher during confinement (24.7%), compared to previous values.
[15]	Sentimental status was significantly associated with the severity of depression symptoms. Those who lived with their partner had better mental health. The severity of the depression varied between regions, with the lowest rates in the southwest and highest in the southeast, northeast, and Wales.
[16]	A large proportion of participants reported being worried about themselves or their family members, feeling isolated during the lockdown, and perceiving the pandemic as a threat to their income.
[17]	An increase in depression symptoms was associated with worsening of memory and other cognitive functions. Increases in significant depressive symptoms in general, but highest among those aged 18–26 years.
[18]	Depressed people had longer sleep duration and waking time in bed. Their diet was more often rich in sugars, and they reported higher alcohol and drug consumption, and higher levels of addiction to television, social media, and gaming.
[19]	The detachment was strongly associated with depressive disorder and associated with a low level of social support.
[20]	Effects of sex, education level, continuity of income, financial situation, and health problems were found in some age groups. Restrictions significantly predicted depressive symptoms in participants aged 18 to 29.
[21]	A total of 14.5% of the adults were at risk for major depressive disorders. The probability of depressive symptoms increased 1.93-fold for each one-unit increase in perceived COVID-19 risk.
[22]	Age was associated with higher resilience, and this was partially mediated by the impact of stressful events on depressive symptoms.
[23]	Significant decrease in diagnoses during first confinement, only slightly offset after 2020. People over 80 were more diagnosed during confinement, except at the end, when younger populations were more diagnosed.
[24]	The psychological impact increased progressively and significantly in the first days of the state of alarm and lockdown, then decreased slightly. In the mental disorder group, being married was associated with a lower risk of depression.
[25]	Higher frequency of suicidal ideation in men during confinement. Most participants felt more relaxed after sports, housework, or talking to someone.
[26]	Significant decrease in the prevalence of major depressive disorders between February 2019 and April 2020. There was no significant change during the six weeks of confinement.
[27]	A total of 12.9% of participants reported clinically significant depressive symptoms. No association was found with gender or previous psychiatric history.
[28]	Depressive symptoms decreased from 2020 to 2021. Pre-pandemic depression diagnosis was associated with higher incidence levels and increased depressive symptoms in 2020 and 2021.
[29]	Progressive increase in depressive symptoms between 2019 and 2020 except in people over 65 and African Americans.
[30]	Increased depression in 26% of participants. No significant associations with sex, age, or mental health problems.
[31]	Psychological symptoms were more frequent and severe in patients with a history of depressive disorder. Frequent feelings of boredom and switching from psychotropic drugs were the main predictors of moderate-severe depression in the patient group.
[32]	No significant association was found between gender and depressive symptoms. Those who reported feeling unwell or very sick were more likely to have depressive symptoms. Respondents who complied with isolation measures were 1.63 times more likely to have depressive symptoms.
[33]	In men, total DASS-21 scores decreased as age increased. Depressive symptoms were highest at the beginning of the pandemic.
[34]	A total of 1.9% of participants met the diagnostic criteria for a major depressive disorder. Depressive symptoms were elevated at the beginning of the pandemic and decreased as the first lockdown progressed, but the prevalence of depressive disorder was higher one year after the start of confinement. The prevalence of depressive symptoms in those with a history of the condition was higher during periods of confinement.
[35]	A significant increase in depressive symptoms was observed one month after the beginning of confinement, without decreasing to baseline levels afterward.
[36]	The prevalence of affective disorders increased by 12.5 during the pandemic. Being screened for COVID-19, regardless of the test result, was associated with increased suicidal tendencies.
[37]	Fluctuation of depressive symptoms when confinement measures decreased.
[38]	In April 2022, cut-off points for clinically relevant depression were greater than 28.3%. Depression levels increased from 21% in April 2020 to 28% in April 2022.
[39]	The prevalence of depressive symptoms increased significantly during the COVID-19 pandemic. Having been quarantined in the 2 weeks before the survey and being a childminder were associated with a relative increase in depressive symptoms. Predictors of depressive symptoms were similar in people with and without a history of mental pathologies.
[40]	There was a statistically significant increase in the prevalence of depressive symptoms at the beginning of confinement, compared to the months in which confinement measures were relaxed and reinstated, without finding differences between the hardest confinement times.
[41]	The year of the pandemic was associated with a significant decrease in depressive symptoms compared to the previous year.
[42]	Mental health was worse at the beginning of the pandemic, but its rates decreased as the year progressed and confinement measures decreased. Those with pre-existing mental health conditions had consistently higher rates of depressive symptoms. In the last wave, the increase in symptoms was greater in those who had no psychiatric history.
[43]	Depressive symptoms maintained the same level from the beginning of confinement. In groups with high-chronic mental disorders, depressive symptoms were slightly but persistently lower during the pandemic than before, with oscillations depending on the severity of confinement. In people with no psychiatric history, depressive symptoms were persistently higher during the pandemic than before it.
[44]	Significant reductions in the diagnosis of depressive disorder were observed during the lockdown period for all sex and age groups.
[45]	Participants with moderate/severe depressive symptoms were found to consume dairy products less frequently, along with fruits, nuts, and vegetables, and more often foods rich in fats and sugars.
[46]	Subjects with an insecure, avoidant, fearful, or anxious attachment style showed an increased risk of depressive symptoms.
[47]	Difficulties in accessing health care were related to a worsening of depressive symptoms. Worsening was more common among those who reported a lack of daily structure or prolonged bedtimes.
[48]	Depressive symptoms increased from the onset of the pandemic to the year of follow-up. People with no history of mental disorders experienced a sharp increase in depressive symptoms from baseline to 12 months, while depressive symptomatology remained stable at a high level in people who received psychiatric/psychological treatment.
[49]	Depressive symptoms remained stable in most waves and periods of confinement. There was a linear increase in symptoms for women but not for men
[50]	Only 20% of participants experienced no symptoms of depression. The prevalence of depressive symptoms among women was 84.4%, and among men 71.2%. People who experienced more than 30 days of lockdown had a significantly higher risk of depression.
[51]	Depressive symptoms remained stable earlier during the lockdown and after confinement. Very weak evidence that the depressed group showed a decrease in depressive symptoms between before and during confinement while the non-depressed group remained stable. The depressed group showed significant decreases in mean sleep duration between before and during confinement, while the group without depression increased mean sleep duration
[52]	A total of 19.2% of participants showed significant psychological distress, with moderate to severe anxiety-depression symptoms being more prevalent in women than in men.
[53]	Compared to before the lockdown, depressive symptoms during the lockdown increased by 132%. Lower likelihood of worsening of symptoms in subjects aged 35 to 54 years. An increase in the size of the municipality was linked to worsening symptoms.
[54]	There were no significant changes in depressive symptoms throughout the lockdown waves. Participants without pre-existing mental health conditions did not change over time.
[55]	The number of participants with depressive symptoms did not differ statistically significantly between the onset of the pandemic and 6 months later.
[56]	A total of 33.7% of the sample reached the level for diagnosis of a major depressive episode. Participants in mental health treatment had significantly higher levels of depression. Of the participants without treatment, 31.4% scored above the cut-off for a diagnosis of depression.
[57]	A significant increase in depressive symptoms was observed during the first weeks of confinement. General psychopathology was associated with worsening relationships with relatives, overeating, fear of gaining weight, use of social media and anti-stress medication, and impaired sleep quality.
[58]	The overall prevalence of depression was 30.13%.
[59]	Results for moderate to severe symptoms were significantly higher than those reported before the pandemic.
[60]	The reported prevalence of depression was around 30.5%. In the third week, the incidence was significantly higher compared to the second week. Depression at week 1 was absent in 76.6% of participants. At week 3, 62.2% had no depressive symptoms.
[61]	The prevalence of depressive symptoms increased between April and May 2020. Statistically significant associations were found between depressive symptoms and psychiatric history. Social support associations did not have a statistically significant effect.
[62]	Participants who self-isolated before lockdown and those who felt very isolated during confinement had higher levels of depression symptoms
[63]	Increases in depression and decreases in quality of life in times of COVID-19 compared to before the pandemic.
[64]	Depressive symptoms increased significantly between April and May 2020. Weeks of exposure to the pandemic and its containment measures were significantly associated with worsening depressive symptoms.
[65]	Significant increase in depressive symptoms between the second and fifth week of confinement.

**Table 2 brainsci-13-00899-t002:** Main characteristics of the studies included in the systematic review. Risk-based and protective factors, as well as the severity of the symptoms, are also shown.

Reference	Country/Region	Population (n) % Female (% Controls If Any)	Risk Factors	Protective Factors	Minimal/Mild Symptoms (%)	Moderate/Severe Symptoms (%)	Age (Median or Mean ± SD)	Depression Evaluation Questionnaire
[13]	England	(36,520) 75.8	Female Younger age Lower education level Lower income Pre-existing mental health conditions	NR	77	22	NR	Patient Health Questionnaire (PHQ)-9
[14]	Italy	(1515) 65.6	Minor Single status Lower income Higher use of social media Fear of leaving home	Use of personal protective equipment Living with partner/s	NR	NR	42	PHQ-2
[15]	United Kingdom	(1006) 54	Younger age Female Lower income Unemployment	Older age Physical activity Living with partner/s	NR	NR	NR	World Health Organization’s Well-Being Index (WHO)-5/PHQ-5/Perceived Stress Scale (PSS)-10
[16]	Nigeria	(966) 49.6	Female Concern about infection	Higher educational level Higher income Living with partner/s Social support	NR	NR	29	NR
[17]	Italy	(1215) 71.1	Female Younger age Teleworking or underemployment	NR	NR	18	43.2 ± 14.5	Self-reporting questionnaire
[18]	Portugal	(968) 78.5 (50)	Health professionals Wake later Consumption of sugar-rich beverages Higher use of social media	Higher educational level Physical activity	NR	NR	50.7 ± 12.7	Online self-reporting questionnaire
[19]	Spain	(3305) 48.7	Female Younger age Loneliness Lower education level Poor physical health Pre-existing mental health conditions Low social support	NR	NR	NR	NR	PHQ-9/Oslo Social Support Scale (OSSS)
[20]	Poland	(115) 50.5	Younger age Female Difficulty in family relationships Lower perceived health Fear and uncertainty	Social support	NR	NR	NR	PHQ-9/self-reporting questionnaires
[21]	South Africa	(957) 73.3	COVID-19 concern Pre-existing mental health conditions Childhood trauma	Higher quality of life	NR	NR	46.3 ± 12.9	General Health Questionnaire (GHQ)-28
[22]	Italy	(21,334) 80.5	Younger age Stressful events	Resilience	NR	NR	38.9 ± 12.8	PHQ-9/PSS
[23]	Germany	(3,021,042) 63.8	Older age	NR	NR	NR	NR	NR
[24]	Spain	(21,207) 69.6	Female Lower income COVID-19 infection or close cases	Advanced age Work autonomy Higher income Living with partner/s Being retired	NR	NR	39.7 ± 14.0	Depression Anxiety Stress Scales (DASS)-21/Impact of Event Scale (IES)
[25]	Libya	(31,557) 65.8	Younger age Female Single status Domestic violence Higher educational level Job loss Lower income COVID-19 infection or close cases	NR	NR	13.6	NR	PHQ-9/IES-Revised (IES-R)
[26]	Ireland	(2061) 51 (49.5)	Younger age Living out of town Less resilience Loneliness More somatic problems	NR	NR	NR	NR	NR
[27]	Greece	(1443) 72.9	Higher educational level Perceived severity COVID-19 concern Married or divorced	NR	NR	NR	NR	Self-reporting questionnaire
[28]	Germany	(12,732) 60.8 (52.6)	Pre-existing mental health conditions	NR	NR	NR	54.2 ± 15.7	PHQ-2/Three-Item UCLA Loneliness Scale (TIL)
[29]	United States	(5075) 51.6	Younger age	NR	NR	NR	47.2 ± 17.6	PHQ-2
[30]	Germany	(396) 70.2	NR	NR	NR	10.4	NR	Online self-reporting questionnaire
[31]	France	(415) 66 (16.6)	Loneliness Boredom Pre-existing mental health conditions Change of pharmacotherapy Going out daily	Physical activity Living with partner/s	NR	NR	38	PHQ-9
[32]	Libya	(10,296) 77.6	Younger age Married Higher educational level Financial problems Current health status Domestic violence Compliance with confinement	Being retired Older age	NR	NR	28.9 ± 8.5	Online self-reporting questionnaire
[33]	Italy	(1401) 35.8	Female	NR	1.12	0.54	44 ± 10.5	DASS-21
[34]	Holland	(3606) 61.9	Female Younger age Pre-existing mental health conditions	NR	NR	NR	57.4 ± 11.9	Online self-reporting questionnaire
[35]	Spain	(5090) 79	MinorLoneliness	NR	NR	NR	NR	Online self-reporting questionnaire
[36]	Czech Republic	(6327) 53 (52.2)	COVID-19 concern Economic concerns	NR	NR	NR	NR	Mini Interna-tional Neuropsychiatric Interview (MINI)/PHQ-9
[37]	France	(296) 74.7	Loneliness Boredom Change of pharmacotherapy	Social support	NR	NR	39	MINI/PHQ-9
[38]	Austria	(1031) 50.3	Female Unemployment Lower education level	Being retired Living with partner/s	NR	NR	NR	PHQ-9/PSS-10
[39]	Holland	(1517) 64.3 (22.1)	Pre-existing mental health conditions Female Loneliness	NR	NR	NR	56.1 ± 13.2	NR
[40]	Italy	(1258) 75.4	Female Job loss COVID-19 concern Younger age	Older age Living with partner/s Perception of security	NR	NR	NR	NR
[41]	Qatar	(6064) 20.9 (83.5)	Female Lower educational level	Older age	NR	NR	NR	PHQ-9
[42]	United Kingdom	(2691) 53.5	Female Minor Pre-existing mental health conditions	NR	NR	NR	NR	PHQ-9
[43]	Holland	(1714) 63.8	NR	NR	NR	NR	56.4 ± 13.0	Quick Inventory of Depressive Symptoms (QIDS)-11/Penn State Worry Questionnaire (PSWQ)
[44]	Spain	(3,640,204) 47	NR	Older age	NR	NR	47	NR
[45]	Italy	(5008) 63	Minor Female Unemployed Students Loneliness	Physical activity	NR	47.5	38	GHQ-12/Center for Epidemiologic Studies Depression Scale (CES-D)/IES-R
[46]	France	(1753) 67.8	Female Economic concerns Homosexual or bisexual	Living with partner/s	11.8	22.5	NR	Major Depression Inventory (MDI)
[47]	Germany	(5135) 59.9 (79.8)	Difficult access to health system during the pandemic Physical inactivity Impact on daily activities	NR	NR	NR	NR	Self-reporting questionnaire
[48]	Germany	(1338) 80.4	Minor Pre-existing mental health conditions Lower educational level Risk group for COVID-19	Older age	NR	NR	NR	PHQ-9/TIL/Stress module of PHQ-9
[49]	Austria	(12,029) 51.5	Younger age Health workers or teleworkers COVID-19 infection or close cases Minor	NR	NR	NR	NR	PHQ-9
[50]	Gulf Cooperation Council States	(14,171) 67.3	Female Minor Pre-existing mental health conditions	Physical activity	80.1	20	NR	PHQ-9/IES-R
[51]	European Countries	(252) 73.8 (52)	NR	NR	NR	NR	NR	PHQ-8
[52]	Chile	(1078) 51.1	Female Loneliness Living in the most affected areas Pre-existing mental health conditions Older age Job loss Lower income	NR	NR	NR	NR	PHQ-4
[53]	Italy	(6003) 50.7	Female High levels of previous outdoor activities Tobacco user	NR	NR	NR	NR	PHQ-9
[54]	United Kingdom	(3067) 55.1	Female Younger age Lower socio-economic level Pre-existing mental health conditions	Older age	NR	26.1	NR	PHQ-9
[55]	Austria	(437) 52.9	NR	NR	NR	NR	NR	PHQ-9
[56]	Argentina	(10,053) 83.4	Younger age Pre-existing mental health conditions Female Lower income	NR	18.5	28.6	41.5 ± 11.5	PHQ-9
[57]	Italy	(671) 71.4 (19.4)	Lower income	Living with partner/s	NR	NR	NR	Brief Symptom Inventory (BSI)/Global Severity Index (GSI)
[58]	Kuwait	(4132) 69.3	Female Single status Job loss Higher risk of loss Pre-existing mental health conditions Younger age Higher BMI Lower educational level Higher use of social media	Being retiredPhysical activity	NR	NR	NR	PHQ-9
[59]	Australia	(13,829) 50.1	Job loss COVID-19 concern Impact on daily activities Female Loneliness Younger age Lower income	NR	26.5	27.6	NR	PHQ-9
[60]	India	(1395) 57.7	Younger age Female Lower educational level Single status	NR	28	1.7	25.0 ± 10.2	DASS-21
[61]	India	(159) 35	Pre-existing mental health conditions	NR	NR	NR	27.4 ± 9.2	PHQ-4
[62]	United Kingdom	(600) 74	Economic concerns	Essential workers Social support	NR	NR	36.7 ± 13.5	Hospital Anxiety and Depression Scale Reference Zigmond and Snaith5 (HADS)/WHO-5/Oxford Capabilities Questionnaire for Mental Health (OXCAP-MH)
[63]	Austria	(1005) 53	Younger age Female Unemployed Lower income	Older age Physical activity	NR	21	NR	PHQ-9
[64]	Italy	(20,720) 71	Female Pre-existing mental health conditions COVID-19 infection or close cases Comorbidities	Higher levels of life satisfaction Living with partner/s	NR	48.9	40.4 ± 14.3	DASS-21/GHQ-12/Suicidal Ideation Attributes Scale (SIDAS)/Severity-of-Acute-Stress-Symptoms-Adult scale (SASS)
[65]	Spain	(1041) 81	Female	NR	NR	NR	39.8 ± 13.5	NR

NR: not reported.

## Data Availability

All data was obtained from the referenced papers and is available through them. All the data utilized for this publication are presented in the tables. There are no further data to be displayed or stored beyond what has been shown in this study.

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
