# Peer review of "Relationship between COVID-19 Pandemic Confinement and Worsening or Onset of Depressive Disorders"

_brainsci, 2023, doi:10.3390/brainsci13060899_

Round 1

Reviewer 1 Report

The study was well conducted. The authors just need to check for proofreading

Further proofreading needed

Author Response

Reviewer 1: The study was well conducted. The authors just need to check for proofreading

We appreciate the reviewer's feedback and kind comments. We have done proofreading of the manuscript.

Reviewer 2 Report

Result:

·         Both tables are quite lengthy.

·         It is preferable to state how many reported protective factors, minimal/mild symptoms, and moderate/severe symptoms,  among the total studies (country).

 Conclusion:

·         In the conclusion mention the important factors that influence the appearance or worsening of depressive symptoms during COVID-19 confinement.  

Author Response

Reviewer 2
Both tables are quite lengthy. It is preferable to state how many reported protective factors, minimal/mild symptoms, and moderate/severe symptoms, among the total studies (country).

We acknowledge the reviewer's comment and kind suggestions. Given that the tables are already quite extensive and that another reviewer has requested additional columns, it is not possible for us to add more columns to the tables. However, this information is indicated in the text (section “Risk factors for depressive symptoms during confinement”).

Conclusion:

  • In the conclusion mention the important factors that influence the appearance or worsening of depressive symptoms during COVID-19 confinement.

 The text added to fix this issue in the conclusion section (highlighted in red).

Reviewer 3 Report

To the authors:

The idea of conducting a review of studies with the objective of analyzing the impact of confinement during the COVID-19 pandemic on depressive symptoms among individuals with and without a history of depression seems of interest to us, especially due to the implications that the results obtained may have in the development of future prevention programs in the field of mental health. However, we believe that this work has some aspects that, in our opinion, should be improved. We will structure our comments according to the different sections of the paper.

Introduction

The introduction is too brief and does not delve into (considering that there are various reviews of previous studies about the repercussions of COVID-19 on mental health) the necessity to delimit the COVID-19-depression relationship. We recommend that the authors provide solid arguments justifying the suitability of the objective of this review.

Materials and Methods

• We suggest that the authors, to facilitate the analysis of the content of Table 1, repeat the header of that table throughout the different sheets in which it is included.

• We believe it necessary to present more data about the characteristics of the samples of the reviewed studies. Specifically:

o   Reference to different aspects of the evaluated population: Does it only include the general population? Are professional groups analyzed (e.g., health professionals)? Is the sample limited to people who have a recent history of recurrent major depressive disorder?

o   Data about the age of the study samples: means, age range... (considering that, as the authors conclude, one of the risk factors is being young)

• Given that the authors consider the severity of depression symptoms according to the scores obtained from different self-reports (see lines 86-87), in Table 1 it is advisable to include the instruments that have been used to evaluate depression in the different studies.

• When discussing the results, in addition to indicating the percentage of studies that report certain risk and protection factors, it should be specified which studies these are. In those cases with a high number of works, some particularly significant example could be provided.

• In the data synthesis subsection (lines 99-100), it is indicated that in this review study data about changes in depression symptoms after COVID-19 confinement are provided, but when discussing the results only the "classification of depressive symptoms during confinement" is referenced. However, it is striking that it is in the discussion section where a table is presented and the main results obtained in the analyzed studies are discussed. Can the authors explain why all the results are not presented in this section?

Discussion

We suggest to the authors a new drafting of the discussion section, carrying out a series of changes that we believe will improve its content. Specifically, we recommend that the authors:

·       Accompany the comments provided in this section with the inclusion of the references of the reviewed studies. For example, in the second paragraph (lines 164-166), it is pointed out that some research concludes that individuals with previous symptoms of depression worsen after COVID-19 confinement, but it does not specify which works those are. Similarly, no reference is made to the studies that report the relationship between the onset of depressive symptoms and life changes related to the pandemic (see lines, 216-219).

·       Provide some explanation for the divergent results obtained in different studies. For example, to what do they attribute that in some works it is reported that in people with a previous diagnosis of depression there is an increase in depressive symptoms during COVID-19, while in others a decrease in symptoms is noted during the confinement period? (see lines 169-172).

·       Expand more in the discussion about the risk and protective factors derived from the review of the studies

·       Explain what they mean by "limited study designs" (line 169).

·       Change Table 2 for the results section and discuss the main results obtained in that section.

·       Eliminate the errors that appear in lines 176-177.

·        Given that the 4th and 5th paragraph of the discussion refer to limitations of the review study (e.g., non-inclusion of research conducted on all continents, use of different instruments to evaluate depressive symptoms in the studies,..), we recommend including a section or paragraphs detailing the main strengths and limitations of this study.

·       Advance what the main practical implications that can be derived from the results obtained from this review of studies are (e.g., proposals for measures to be adopted in people with high levels of previous depressive symptoms, in older people living alone, …).

Author Response

The idea of conducting a review of studies with the objective of analyzing the impact of confinement during the COVID-19 pandemic on depressive symptoms among individuals with and without a history of depression seems of interest to us, especially due to the implications that the results obtained may have in the development of future prevention programs in the field of mental health. However, we believe that this work has some aspects that, in our opinion, should be improved. We will structure our comments according to the different sections of the paper.

Introduction

The introduction is too brief and does not delve into (considering that there are various reviews of previous studies about the repercussions of COVID-19 on mental health) the necessity to delimit the COVID-19-depression relationship. We recommend that the authors provide solid arguments justifying the suitability of the objective of this review.

Thank you for your time and for your kind comments. We sincerely thank the reviewer for their valuable input and constructive feedback. The appropriate citations and references have been added to address this issue. The new text is highlighted in red.

Materials and Methods

  • We suggest that the authors, to facilitate the analysis of the content of Table 1, repeat the header of that table throughout the different sheets in which it is included.

We have done it for both tables.

  • We believe it necessary to present more data about the characteristics of the samples of the reviewed studies. Specifically:
  • Reference to different aspects of the evaluated population: Does it only include the general population? Are professional groups analyzed (e.g., health professionals)? Is the sample limited to people who have a recent history of recurrent major depressive disorder?

This information is in Materials and methods (Excluding/including criteria). This work only included articles focused on the general population and excluded those that specifically emphasized a particular group such as healthcare workers.  We also indicated in this section that we included studies that evaluated only individuals with a history of mental disorders, as well as others that mixed individuals with a history of mental disorders with those without history.

  • Data about the age of the study samples: means, age range... (considering that, as the authors conclude, one of the risk factors is being young)

Added to table 1 (new column)

  • Given that the authors consider the severity of depression symptoms according to the scores obtained from different self-reports (see lines 86-87), in Table 1 it is advisable to include the instruments that have been used to evaluate depression in the different studies.

Added to table 1 (now table 2, new column)

  • When discussing the results, in addition to indicating the percentage of studies that report certain risk and protection factors, it should be specified which studies these are. In those cases with a high number of works, some particularly significant example could be provided.

The number of works and percentages are indicated at “Protective factors for depressive symptoms during confinement” section (page 15).

  • In the data synthesis subsection (lines 99-100), it is indicated that in this review study data about changes in depression symptoms after COVID-19 confinement are provided, but when discussing the results only the "classification of depressive symptoms during confinement" is referenced. However, it is striking that it is in the discussion section where a table is presented and the main results obtained in the analyzed studies are discussed. Can the authors explain why all the results are not presented in this section?

Table 2 has been moved to the results section. In any case, the data shown in this table are different from those extracted from the papers in the systematic review. Table 1 shows the data obtained regarding the items analyzed in this paper, while Table 2 shows the general conclusions from the papers reviewed. Anyway, both tables are now in results, and the results described and discussed. Now Table 1 is Table 2 and vice versa, as it seems to us a more logical order for this organization.

Discussion

We suggest to the authors a new drafting of the discussion section, carrying out a series of changes that we believe will improve its content. Specifically, we recommend that the authors:

  • Accompany the comments provided in this section with the inclusion of the references of the reviewed studies. For example, in the second paragraph (lines 164-166), it is pointed out that some research concludes that individuals with previous symptoms of depression worsen after COVID-19 confinement, but it does not specify which works those are. Similarly, no reference is made to the studies that report the relationship between the onset of depressive symptoms and life changes related to the pandemic (see lines, 216-219).

The appropriate references have been added.

  • Provide some explanation for the divergent results obtained in different studies. For example, to what do they attribute that in some works it is reported that in people with a previous diagnosis of depression there is an increase in depressive symptoms during COVID-19, while in others a decrease in symptoms is noted during the confinement period? (see lines 169-172).

This explanation has been added to the “Strengths, limitations, and perspectives”:

Note that divergent results from different studies are described in this work. For example, some papers report an increase in depressive symptoms during COVID-19 in people with a previous diagnosis of depression, while others report a decrease in symptoms during the period of institutionalization. The divergent results obtained in different studies about depression and confinement during the COVID-19 pandemic can be attributed to several factors, including variations in study design, sample characteristics, measurement tools, and contextual factors. Cross-sectional studies provide a snapshot of a particular point in time, whereas longitudinal studies are better suited to capture the dynamic nature of mental health during the pandemic. Differences in participant characteristics may also contribute to divergent results, as noted in the results of this study. Factors such as age, gender, socioeconomic status, pre-existing mental health conditions, and access to resources may influence the impact of confinement on depressive symptoms. The assessment tools used to measure depression and related symptoms can differ across studies. Variations in the choice of validated scales, self-report measures, clinical interviews, or diagnostic criteria can lead to different interpretations and results. Differences in the sensitivity and specificity of these instruments can affect the identification and reporting of depressive symptoms. The specific timing and duration of the confinement period can influence the findings. Studies conducted at different stages of the pandemic or during varying lengths of confinement may capture different phases of the mental health impact. Initial phases of the pandemic may have induced more anxiety and uncertainty, while prolonged confinement may have led to feelings of isolation and reduced social support.

It is important to consider these factors when interpreting divergent results. A comprehensive understanding of the complex interactions between individual, social, and environmental factors is crucial to fully comprehend the impact of the pandemic on mental health outcomes like depression.

  • Expand more in the discussion about the risk and protective factors derived from the review of the studies.

We have discussed these results in lines 229 to 278.

  • Explain what they mean by "limited study designs" (line 169).

We added the sentence “regarding the absence of cohort studies with control groups that would allow for comparisons regarding exposure to risk factors and counteracting findings of other studies.” (red letters)

  • Change Table 2 for the results section and discuss the main results obtained in that section.

The table has been changed. These results are summarised in results section (lines 128 to 186)

  • Eliminate the errors that appear in lines 176-177.

The errors are only visible in the PDF version of the manuscript and involve two references. We will make the necessary changes to ensure that it does not happen in the new version.

  • Given that the 4th and 5th paragraph of the discussion refer to limitations of the review study (e.g., non-inclusion of research conducted on all continents, use of different instruments to evaluate depressive symptoms in the studies,..), we recommend including a section or paragraphs detailing the main strengths and limitations of this study.

Section “Strengths, limitations, and perspectives” has been added to cover this. (page 17).

  • Advance what the main practical implications that can be derived from the results obtained from this review of studies are (e.g., proposals for measures to be adopted in people with high levels of previous depressive symptoms, in older people living alone, …).

Section “Strengths, limitations, and perspectives” has been added to cover this. (page 17).

Round 2

Reviewer 3 Report

The authors have made all the modifications that have been suggested. The final result significantly improves the manuscript and I recommend it for publication. Congratulations to the authors.